# Mitigating Lithium Dissolution and Polysulfide Shuttle Effect Phenomena Using a Polymer Composite Layer Coating on the Anode in Lithium–Sulfur Batteries

**DOI:** 10.3390/polym14204359

**Published:** 2022-10-16

**Authors:** Hyukmin Kweon, William Kim-Shoemaker

**Affiliations:** 1Civil and Environmental Engineering, University of California, Los Angeles, CA 90095, USA; 2BenSci Inc., 2321W 10th Street, Los Angeles, CA 90006, USA; 3Jericho High School, 99 Cedar Swamp Rd., Jericho, NY 11753, USA

**Keywords:** lithium sulfur battery, polysulfide, shuttle effect, dendrite, polyaniline, graphite, COMSOL

## Abstract

To mitigate lithium dissolution and polysulfide shuttle effect phenomena in high-energy lithium sulfur batteries (LISBs), a conductive, flexible, and easily modified polymer composite layer was applied on the anode. The polymer composite layer included polyaniline and functionalized graphite. The electrochemical behavior of LISBs was studied by galvanostatic charge/discharge tests from 1.7 to 2.8 V up to 90 cycles and via COMSOL Multiphysics simulation software. No apparent overcharge occurred during the charge state, which suggests that the shuttle effect of polysulfides was effectively prevented. The COMSOL Multiphysics simulation provided a venue for optimal prediction of the ideal concentration and properties of the polymer composite layer to be used in the LISBs. The testing and simulation results determined that the polymer composite layer diminished the amount of lithium polysulfide species and decreased the amount of dissolved lithium ions in the LISBs. In addition, the charge/discharge rate of up to 2.0 C with a cycle life of 90 cycles was achieved. The knowledge acquired in this study was important not only for the design of efficient new electrode materials, but also for understanding the effect of the polymer composite layer on the electrochemical cycle stability.

## 1. Introduction

Recently, lithium-ion batteries (LIBs) have achieved great success and are widely used in electric vehicles, consumer electronics, and stationary energy storage systems [1]. Many studies have been conducted to achieve highly safe, high-energy-density storage systems with sustainable electrochemical performance [2,3,4,5,6]. The focus of most studies has been LIB modifications that improve traits such as lifespan, efficiency, and size. Highly reactive electrode/electrolyte materials provide increased power and performance, but result in fire and/or explosion and accelerated degradation even when the battery is not used [7]. In addition, improved LIBs can hardly support the growing demand for high-energy-density electrochemical cells. To overcome these limitations, lithium–sulfur batteries (LISBs) have been proposed as a potential alternative to current state-of-the-art LIBs due to their theoretical high capacity (1675 mAh/g) and energy density (2510 Wh/kg) [8,9]. Sulfur is also considered a sustainable resource due to the low environmental impact of its harvest and the possibility of reusing sulfur from used batteries [10]. Table 1 summarizes main milestones of the development in LISB research.

Despite considerable research on the LISBs, their practical implementation is hinder by the challenge of a dramatically shortened cycle life. This is mainly due to the polysulfide shuttle and lithium dissolution effects, which lead to the formation of dendrites on the lithium anodes as lithium ions return to the anode; these ions them accumulate on cathode as polysulfide species [29,30]. Together, the non-dissolvable intermediate lithium polysulfides on the sulfur cathode and uncontrollable growth of lithium dendrites on the anode surface reduce the activity of LISBs [31]. Several approaches have been developed to address these issues over the past few decades including cathode design, separator modification, use of novel electrolytes, and anode improvement [32]. To fabricate advanced sulfur-based composite cathodes, various porous materials and conductive materials such as porous carbon material [33] and graphene-based material [34] are considered for their high electrical conductivity. A separator, usually a polymer membrane, acts as an electron insulator to prevent short circuits. Modified separators have proven to be an efficient way to inhibit polysulfide shuttles [32]. In addition, anode improvements in LISBs deserve attention from researchers. Deactivation of lithium anodes is the most common reason for failure because lithium is highly reactive to organic electrolytes and form solid electrolyte interfaces. Lithium ions dendrites form that are deposited irregularly on the lithium anode while the material is displaced.

In this study, we propose that a polymer composite layer coating on anode in LISBs could mitigate lithium dissolution and polysulfide shuttle effect phenomena. The polymer composite layer was composed of a conductive, flexible, and easily modified material (polyaniline; PANi, emeraldine base, Mw = ~20,000 g/mol with functionalized graphite) and was applied to the surface of the pre-lithiated graphite and carbon black anode. This coating layer could withstand the volume change of lithium during the cycle and enhance the cycling ability of LISBs. In addition, the polymer composite coating materials are chemically stable enough not to dissolve in the electrolyte, and are significantly conductive [35,36]. The applied conductive polymer composite material improved the cycle life of the LISBs. Additionally, the ionic conductivity of the polymer composite layer was enhanced by a doping treatment with hydrofluoric acid (HF) [37]. Hence, the doped layer can assist in the structural maintenance of the anode while preserving conductivity traits of the original relationship between the electrolyte and the anode.

## 2. Materials and Methods

### 2.1. Materials

Polyvinylidene fluoride (PVDF), n-methyl-2-pyrrolidone (NMP), carbon black, conductive acetylene black nano powder, graphite (TIMCAL TIMREX^®^ KS6), Lithium nickel manganese cobalt oxides (NMC 424, LiNi_0.4_Mn_0.2_Co_0.4_O_2_), sulfur-carbon composite, copper foil (9 µm thickness), and aluminum foil (15 µm thickness) were purchased from MSE Supplies LLC, Tucson, AZ, USA. Lithium perchlorate (LiClO_4_), sulfolane, lithium polysulfide (Li_2_S_8_), sulfuric acid (H_2_SO_4_), potassium permanganate (KMnO_4_), hydrogen peroxide (H_2_O_2_), hydrochloric acid (HCl), and HF were purchased from Sigma-Aldrich. All chemicals were used as received without any treatment or purification.

### 2.2. Material Synthesis

First, a PVDF solution of was prepared in a mass ratio of 1:15 of PVDF to NMP and heated for 12 h. Then, the NMC, sulfur, and carbon black powders were ground together for the cathode in accordance with a mass ratio of 35:60:5 under continuous stirring for 2 h while heated to 155 °C. The slurry was then distributed evenly over the aluminum foil with a doctor blade and dried at 60 °C for 12 h. Next, NMC and graphite were dispersed for the anode in accordance with a mass ratio of 88:12 in a ball-milling machine. After that, the mixture was added to the PVDF solution through sonication. In this method, the degree of pre-lithiation is easily controlled by adjusting the weight ratio of the anode material and the Li metal. The slurry was cast onto a cupper foil with a doctor blade and dried at 60 °C for 24 h. The PANi composite with functionalized graphite was prepared using method by described in a previous work [38,39,40]. To functionalize the graphite, two different oxidation levels (partially and highly) were applied by following procedures. For the partially oxidized graphite, 10 g of graphite was added to 460 mL of concentrated H_2_SO_4_. After the graphite was dispersed in concentrated H_2_SO_4_, 60 g of KMnO_4_ was added. The temperature of the mixed solution was kept below 12 °C, while the reaction was held for 2 h. Then, 960 mL of DI water was added dropwise for 1 h while maintaining the temperature at not more than 45 °C. Subsequently, the reaction mixture was diluted with DI water and quenched with H_2_O_2_. For highly oxidated graphite, additional 300 g of KMnO_4_ was added into the mixed solution gradually, keeping the reaction temperature below 10 °C. The reaction mixture was stirred for 4 days, and then the reaction was terminated by pouring it into a large amount ice before being quenched with H_2_O_2_. The mixtures were purified by centrifugation using HCl and DI water, followed by dialysis. A controlled amount of graphite oxide was dispersed in 60 mL of NMP using ultrasonication for 30 min. Additionally, then different masses (50–250 mg) of PANi was added to the GO suspension while stirring. Following 10 h of constant stirring, the resulting solution was filtered using a 5 µm syringe filter. The prepared polymer composite solution was spin-coated to form a thin polymer film on the anode. Electrolytes were developed by dissolving 1 M of LiClO_4_ in sulfolane. Additionally, lithium polysulfide solutions were added within the electrolyte solution.

### 2.3. Electrochemical Measurements and Characterization

Coin cells (2032) were assembled in an argon-filled glovebox with moisture and oxygen contents below 3 ppm. The quantity of electrolyte was controlled at 12~15 μL per 1 mg sulfur. Galvanostatic charge/discharge tests were carried out using a LANHE battery tester (Wuhan LAND Electronic Co., Ltd., Wuhan, China) within a voltage window of 1.7~2.8 V for up to 90 cycles. Initially, the cells were activated by discharging at a constant current of 0.1 C (1.0 C = 1675 mAh/g) to 1.7 V, and then charged at a constant current of 0.1 C to 2.8 V for three cycles. After activation, the cells were tested at 0.1 C, 0.2 C, 0.4 C, 0.6 C, 0.8 C, 1.0 C and 2.0 C, respectively. The morphologies of the electrode and coated layers were examined with a Phenom Pharos desktop field emission scanning electron microscope (FE-SEM) using a secondary electron detector (SED) from Thermo Scientific (Waltham, MA, USA). To observe electrodes after 90 cycles were disassembled in an argon-filled glove box. All FE-SEM images were captured with 10 kV acceleration voltage.

### 2.4. COMSOL Multiphysics Simulation

One-dimensional (1D) battery simulation comprises four sections: the negative electrode, polymer composite layer, separator, and the positive electrode. This model was modified from an existing implementation of a COMSOL 1D LIB model (COMSOL, Inc., Burlington, MA, USA, Application ID: 686). It includes an isothermal system that models electronic current conduction in the electrodes, ion transport across the battery, material transport in the electrolyte, and Butler-Volmer electrode kinetics in combination with the Nernst equation (assuming law of mass action) and using experimentally measured discharge curves for the equilibrium potential. Table 2 summarizes all the input values required by COMSOL for solving the equations pertaining to the ion transport across the LISB. The other model equations can be found from LISB model (Application ID: 80721). The model comes with default properties associated with the materials used to constructed it. In our model, the negative and positive electrodes have a volumetric fraction of $\varepsilon_{l}$ = 0.264 and $\varepsilon_{l}$ = 0.357, respectively. An electrical conductivity of $\sigma_{l}$ = 10–100 S/m dependent on material composition ratio and oxidation level. The electrolyte has a diffusion constant of 7.4 × 10^−11^ m^2^/s.

A schematic diagram of the 1D LISB model as it appears in the COMSOL program can be seen in Figure 1. The model is split into four sections, each having different thicknesses: the negative electrode (25 µm without polymer layer), polymer composite layer (5 µm), the separator (0 µm, represented as a point), and the positive electrode (20 µm). The model comes with default properties associated with the materials used to construct it. The electrical conductivity of metals was assumed to be effective and to constantly account for the porous nature of the matrix. The diffusion coefficient was set to 1× 10^−9^ m^2^/s.

To create and plot the cell voltage data, cell voltage was selected in the 1D plot group setting and the plot chosen. To find all species concentrations, the default plot was modified to show the concentrations for the last saved time, and for each C-rate individually. Computing the results from the study produced results for lithium concentration as well as the polysulfide volume fraction in the electrolyte, which were graphed and plotted with the provided COMSOL functions.

## 3. Results and Discussion

A pure lithium metal is an ideal anode material; however, the significant volume change due to the dendritic growth can lead to the pulverization of the anode and the expansion of the cell case, which may cause sudden failure and a serious safety hazard [41,42]. To avoid the cell expansion, use of non-lithium anode is an alternative approach out of the primary dilemma of lithium metal anode. Various lithium sources such as lithium chloride (LiCl), lithium hydroxide (LiOH), lithium cobalt (LiCo), lithium oxide (LiO), and lithium metal were used as additives for a passive pre-lithiation to improve initial coulombic efficiency. In this study, pre-lithiated graphite and carbon black anode powder additives were utilized, which results in a high rate capacity and low cost.

A fixed amount of lithium composite materials was applied and deposited onto the metal substrates using the doctor blade (20 µm thickness), then a subsequent polymer composite material coat was added using a spin coater. Figure 2 illustrates the morphology differences between the anodes without and with the polymer composite layer. The pristine pre-lithiated graphite and carbon black anode in Figure 2a has a bumpy surface with irregular thickness, whereas the polymer composite coated pristine pre-lithiated graphite and carbon black anode in Figure 2b has dense and flat surface. As shown in Figure 2b, notably thin coatings (less than 5 µm) are necessary to ensure the benefit of higher energy storage capacity of the lithium composite anode [43].

X-ray photoelectron spectroscopy (XPS) was utilized to further confirm and characterize the polymer coating layer on the surface of the electrode. In the C1s peaks of the pristine pre-lithiated graphite and carbon black anode (Figure 3a), three peaks were located at 284.5, 286.7, and 288.5 eV, which correspond to the carbon-carbon (C=C) bond from graphite and a carbon-oxygen (C-O) and C=O bonds of lithium carbonates, respectively. In the case of the polymer composite coated anode (Figure 3b), an analysis of the C1s orbital energies provided evidence of an amide functional group in polyaniline composite coating layer; carbon-nitrogen (C–N) appears at 285.7 eV and isocyanic acid (HN-C=O) appears at 287.9 eV, respectively. In addition, a new peak appeared at 289.8 eV, which is attributed to the carbon-fluorine (C-F) bonds from the HF doping of the polymer composite coating layer. After coating with polymer composite material, the peak of C=C at 284.5 eV was relatively small due to the formation of a new polymer composite layer. The SEM and XPS results support the successful formation of a new polymer composite layer on the pristine pre-lithiated graphite and carbon black anode.

The galvanostatic charge/discharge tests were carried out to measure the electrochemical performance and profiles from 1.7 to 2.8 V for up to 90 cycles. The galvanostatic discharge curves of the pre-lithiated graphite and carbon black anode without and with the polymer composite layer at various currents (0.1 C, 0.2 C, 0.4 C, 0.6 C, 0.8 C, 1.0 C and 2.0 C) are shown in Figure 4a,b. The reversible discharge capacity without the polymer composite layer can reach up to 1368 mAh/g at 0.2 C. With an increase in current density, the specific capacity gradually decreases to 1266 mAh/g (0.4 C), 1182 mAh/g (0.6 C), 1108 mAh/g (0.8 C), 1012 mAh/g (1.0 C) and 608 mAh/g (2.0 C), respectively (Figure 4a). On the other hand, Figure 4 (b) shows that the reversible discharge capacity with the polymer composite layer can reach up to 1308 mAh/g at 0.2 C. As the current density is increased, the specific capacity gradually decreases to 1230 mAh/g (0.4 C), 1160 mAh/g (0.6 C), 1082 mAh/g (0.8 C), 998 mAh/g (1.0 C) and 580 mAh/g (2.0 C), respectively. No apparent overcharge occurred during the charge state, which suggests that the shuttle effect of polysulfides was effectively prevented. Once the current density goes back to 0.2 C, the specific capacity without the polymer composite layer recovers from 1368 mAh/g to 1352 mAh/g, whereas the specific capacity with the polymer composite layer recovers from 1308 mAh/g to 1302 mAh/g, respectively. These results demonstrate the outstanding rate performance of the pre-lithiated graphite and carbon black anode with the polymer composite layer electrode. Figure 4c presents the rate performance of two electrodes. Note that the discharge capacity clearly decreases during the first a couple of cycles before reaching a steady state. Both electrodes exhibit stable behaviors from 0.2 C to 2.0 C up to 90 cycles, maintaining their initial capacities. The capacity of the high-potential plateau decreases slightly with the increased rate for 2.0 C and 1.0 C, but remains nearly constant for 0.8 C, 0.6 C, 0.4 C, and 0.2 C.

The 1D COMSOL model was built to further evaluate the role of a polymer composite layer in the proposed LISBs. The first parameter was the diffusion coefficient of the electrode without and with the polymer composite layer. The diffusion coefficient did have a significant effect on discharge curves. The discharge curve presents the voltage discharged for the capacity remaining in the LISBs. As shown in Figure 5, each curve corresponds to battery discharge according to different current densities from 0.2 C to 2.0 C. Whether there is the polymer composite layer or not, the discharge curves were not affected at the higher diffusion coefficients. However, lower diffusion coefficient could not output as much voltage at higher currents due to fewer ions crossing the separator at the low diffusion coefficient. The similar predicted discharge curves indicate that the presence of the polymer composite layer does not affect the discharge or energy production of the LISBs.

As reported in the literature, the solubility of polysulfide is one of the capacity fading factors which are the main cause of the current low energy capacity of LISBs [44,45]. This solubility causes a polysulfide shuttle phenomenon that (1) delays completion of a charging process, (2) reduces utilization of an active material in a discharging process, and (3) contributes to capacity fading [46]. One more significant factor of capacity fading is the irreversible precipitate (for example, lithium sulfide, Li_2_S) on the cathode, which is insoluble and electrochemically inaccessible. Therefore, the lithium sulfide was evaluated to determine whether the polymer composite layer on the anode affects the dissolution occurring. Figure 6 represents the volume of lithium sulfide species in the 1D LISBs. Without the polymer composite layer (Figure 6a), a high-volume fraction (0.37) of harmful lithium sulfide can be observed in the electrolyte. However, with the polymer composite layer present, a lower volume fraction (0.34) of the harmful lithium sulfide in the electrolyte (Figure 6b) can be seen. This indicates that the polymer composite layer has a positive effect of generating smaller volume of the lithium sulfide in the LISBs. The reduced lithium sulfide content in the battery system will reduce the polysulfide shuttle phenomenon effect in the LISBs.

Although the effect of the polymer composite layer on the discharge curve and volume fraction of lithium sulfide was confirmed, it is necessary to validate the effect of the polymer composite layer on the anode. Therefore, lithium concentration on both electrodes was evaluated. It was found that there was a lithium peak of 9500 mol/m^3^ on the anode without the polymer composite layer (Figure 7a). With the polymer composite layer, the lithium peak was 14,000 mol/m^3^ on the anode (Figure 7b). This suggests that the polymer composite layer suppresses the dissolution of lithium ions in the electrolyte. In summary, the polymer composite layer diminishes not only the amount of polysulfide species on cathode, but also decreases the amount of dissolved lithium ions on the anode. This could be achieved without jeopardizing the total energy to be produced from the LISBs.

## 4. Conclusions

A conductive, flexible, and easily modified polymer composite layer was proposed to mitigate lithium dissolution and the polysulfide shuttle effect phenomena for LISBs. The electrochemical behavior of LISBs was studied by galvanostatic charge/discharge tests and COMSOL Multiphysics simulation. The charge/discharge rate of up to 2.0 C with a cycle life of 90 cycles can be achieved. Additionally, overcharge was not observed during the charge state, which means that the shuttle effect of polysulfides was effectively avoided. Additionally, the developed COMSOL Multiphysics simulation provides a venue for optimally predicting the ideal concentration and properties of the polymer composite material layer used in LISBs. The polymer composite layer diminishes not only the amount of lithium from anode to electrolyte, but also decreases the amount of lithium polysulfide generation on the cathode. The reduced lithium polysulfide content in the battery system will lower the polysulfide shuttle phenomenon effect in the LISBs resulting increasing the likelihood of achieving high-energy-density LISBs. The LISB knowledge acquired in this study contributes to the tremendous potential for battery innovative designs as storage systems for electric vehicle projects or those that utilize renewable energies such as solar, wind and wave power.

## Figures and Tables

**Figure 1 polymers-14-04359-f001:**
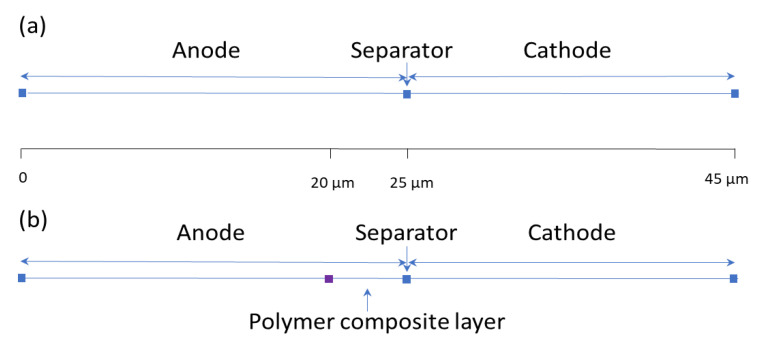
A schematic diagram of the 1D LISBs as they appear on COMSOL: (**a**) Without polymer composite layer; (**b**) With polymer composite layer (purple square).

**Figure 2 polymers-14-04359-f002:**
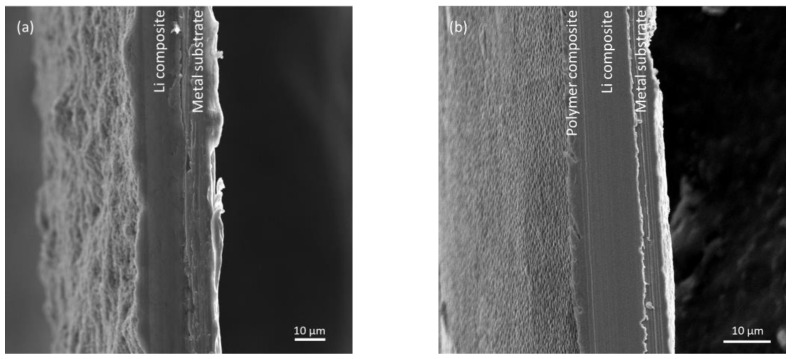
Cross sectional SEM images of anodes: (**a**) Pristine pre-lithiated graphite and carbon black anode; (**b**) Polymer composite coated on the pre-lithiated graphite and carbon black anode.

**Figure 3 polymers-14-04359-f003:**
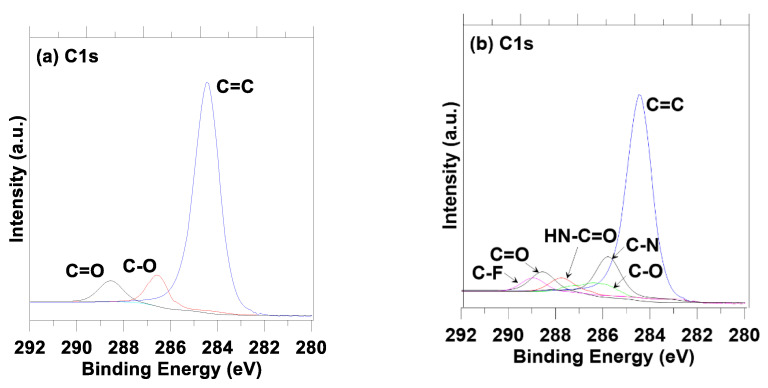
XPS spectra of the following: (**a**) Pristine pre-lithiated graphite and carbon black anode; (**b**) Polymer composite coated pre-lithiated graphite and carbon black anode.

**Figure 4 polymers-14-04359-f004:**
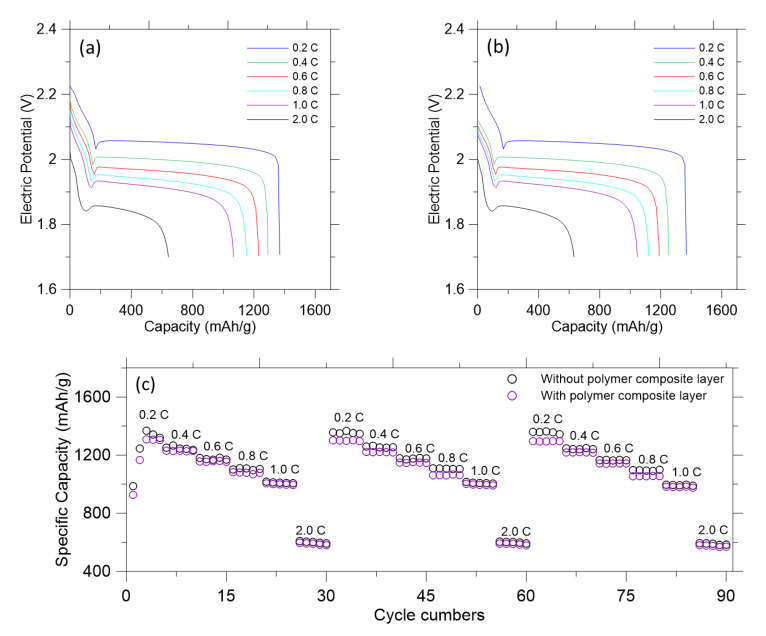
Electrochemical experimental performance of the LISBs. Initial galvanostatic discharge profiles: (**a**) Without the polymer composite layer; (**b**) With the polymer composite layer. (**c**) Rate performance at current density from 0.2 C to 2 C without and with the polymer composite layer.

**Figure 5 polymers-14-04359-f005:**
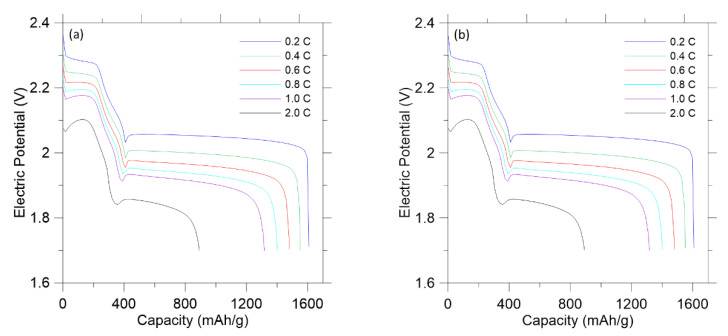
Discharge curves of LISBs at densities varying from 0.2 C to 2.0 C from the COMSOL model: (**a**) Without the polymer composite layer; (**b**) With the polymer composite layer.

**Figure 6 polymers-14-04359-f006:**
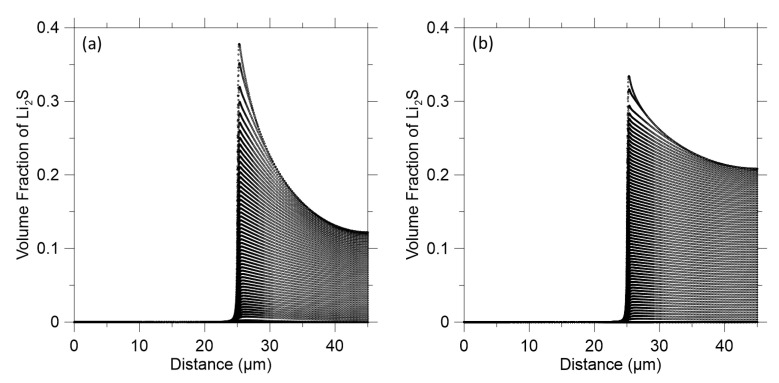
Lithium polysulfide volume fraction on the cathode: (**a**) Without the polymer composite layer; (**b**) With the polymer composite layer.

**Figure 7 polymers-14-04359-f007:**
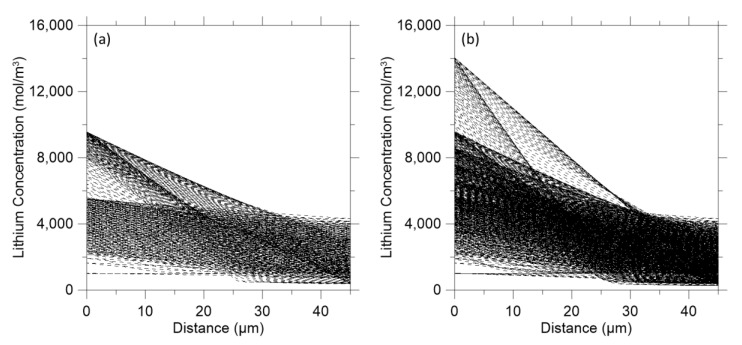
Distribution of lithium concentration on both electrodes: (**a**) Without the polymer composite layer on the anode; (**b**) With the polymer composite layer on the anode.

**Table 1 polymers-14-04359-t001:** Main milestones of the development in LISB research.

Year	Main Milestones	Challenges/Limitations	Approaches	References
1960–2000s	Lithium sulfur cell proposed Solid-state LISB proposed Dioxolane-rich electrolyte for LISBs	Understand the reversible redox reaction of lithium polysulfide Low-rate capability due to its poorly conducting electrolyte	Dioxolane-based electrolytes have been evaluated	[11,12]
2002	Carbon/sulfur cathode was developed Polysulfide shuttle effect was issued	Prolong the residence of polysulfides Demand on the high specific energy and low-cost LISBs	Carbon/sulfur composite cathodes were proposed Carbon nanotube, graphene, and CNT/graphene composite were utilized	[13,14]
2004–2010	Lithium-polysulfide-based electrolyte for LISBs LiNO_3_ as critical electrolyte was proposed Excellent lithium plating/stripping properties were obtained	Lithium dendrite growth Polysulfide shuttle Facilitate the formation of a protective solid electrolyte interphase on the lithium metal anode	Solid electrolyte, and surface modification of lithium were widely studied LiNO_3_ as critical electrolyte additive applied to protect solid electrolyte interphase	[15,16,17]
2013–2017	Deep understanding of polysulfide chemistry High concentration liquid electrolyte Li anode protection	Overall understanding of polysulfide chemistry Further improvements in cell performance.	A slight excess in lithium metal to protect lithium anode Ultrathin lithium foil (<50 µm) is employed	[18,19,20,21]
2018–present	All solid-state LISBs are widely studied Unique polysulfide chemistry was proposed	Polysulfide shuttle Dendritic growth issues Demand for high-performance LISBs Battery safety issues such as the potential leakage risk of liquid electrolytes	Inorganic glassy ceramics and ceramics, organic polymers, and inorganic–organic hybrid electrolytes have been applied Use of organic liquid electrolytes All-solid-state LISBs	[22,23,24,25,26,27]
Future		Demand on the high-energy-density LISBs A large-scale system	High sulfur loading Well-designed electrodes All solid-state lithium-sulfur batteries	[28]

**Table 2 polymers-14-04359-t002:** List of inputs that are required by COMSOL for simulating the ion transport across the LISB.

Variable	Units	Definition
F	C/mol	Faraday’s constant
R	J/(mol K)	Universal gas constant
T	K	Temperature of the simulation
$\varepsilon_{l}$	-	Volumetric fraction of the electrolyte phase
$\sigma_{l}$	S/m	Intrinsic electrolyte conductivity (tensor)
$D_{l}$	m^2^/s	Intrinsic electrolyte diffusivity (scalar)
$t_{l}$	-	Transference number for ions
$f(\varepsilon_{l}$)	-	Ratio between the effective and intrinsic electrolyte properties (diffusivity and conductivity)

## Data Availability

Not applicable.

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
