# Peer review of "Mitigating Lithium Dissolution and Polysulfide Shuttle Effect Phenomena Using a Polymer Composite Layer Coating on the Anode in Lithium–Sulfur Batteries"

_polymers, 2022, doi:10.3390/polym14204359_

Round 1

Reviewer 1 Report

The reported work in this manuscript deals with an innovative approach to solve one of the rather well known issues pertaining to the long term development of rechargeable sulfur batteries (eg dendrites formation and polysulfide formation during use).

The paper is well writtten and the approach is scientific sound. However to better justify the rationale of this work, I see however some room for imprivement of this original manuscript:

1.I suggest to reinforce the bibliographical review and introduce a new table presenting chronological breakthroughs (as well as potential limitations) towards the resolution of the mentioned issues being considered in this paper:

Beyond the use of cited references 21 to 25, I believe that other references might be useful for such a purpose, such as:

Kolosnitsyn and E.V. Karaseva: Lithium sulfur batteries: problems and solutions, Russian Journal of Electrochemistry, 2008, vol 44, n°5, 506-509

S.H. Yu et al, Elecrrochemica Acta, 2021, 9, 10231-10239*

C. Zech et al, J. Mater. Chem., 9, 10231-10239

L. Dong et al, ACS nano 2019, 13, 12, 14172-14181

and likely some others.

2. Just notice also that a brand new scientific book has just been published (June 2022) as regard Li-sulfur batteries by Elsevier, which may also be helpful:

3. Also potentially interesting for the reader of you paper, it might be useful (if accessible) to indicate what is the practical solution currently in place in existing Li-S already placed on the market (since my undeerstanding that altough this is still an emerging market, LI-S batteries implementation is fastly growing for a year or two...

4 (editorial)

page 2, just below table 1: "...is hinder..." shall be written "...is hindered..." in my view ?

Author Response

Dear Reviewer,

We would like to thank the reviewer and the editor for the positive and constructive comments and suggestions of our manuscript entitled “Mitigating Lithium Dissolution and Polysulfide Shuttle Effect Phenomena using a Polymer Composite Layer Coating on Anode in Lithium–Sulfur Batteries”. Those comments are very helpful and valuable to revise and improve our manuscript. We have studied the reviewer’s comments carefully and have made revisions which are marked up using the “Track Changes” function. We have tried our best to revise our manuscript according to the comments and offered specific point-by-point responses to the reviewer’s comments below. We hope that our revision and responses convince you to accept our manuscript for publication.

Reviewer 2 Report

  The authors tried to discuss a scientific problem on ' Mitigating Lithium Dissolution and Polysulfide Shuttle Effect Phenomena' for lithium-sulfur batteries. However, I think the whole research is organized arbitrarily, and the data seems to be of poor quality, especially for the part on electrochemical performance, and it is hard to consider the simulation by COSMOL in a sound chemical background. Therefore I suggest that the current results and discussion can hardly meet the publication requirement for any scientific papers. I suggest the authors establish additional experiments and compare them with this research on lithium-sulfur batteries.

 Some other suggestions are listed below.

1  Table 1 shows a comparison of key features of lithium-ion and lithium-sulfur batteries, and I think it is unnecessary to do such work as the focus of research is limited to lithium-sulfur batteries. The authors should pay more attention to the strategies to improve the performance of lithium-sulfur batteries. As I know, hundreds of cycles for such batteries have been often observed in recent literature.

2 The authors adopt NMC 424 to mix with sulfur, which is a usual operation, and thus the authors should fully explain their design. In contrast, the main researchers mix sulfur with various functional carbon.

3 The authors adopt a homemade electrolyte, LiClO4/sulfolane, and such experiment design is arbitrary, as others usually adopt carbonate or ether electrolyte with additives such as LiNiO3. If limited by the homemade electrolyte, the authors should have little chance to make good research.

Author Response

(The authors gave the same response as above.)

Reviewer 3 Report

In the manuscript, a conductive, flexible, and easily modified polymer composite layer was applied on the anode, in order To mitigate lithium dissolution and polysulfide shuttle effect phenomena in high energy lithium sulfur batteries (LISBs), the results may contribute to the design of efficient new electrode materials. More explanation about COMSOL Multiphysics simulation should be given, which may make the manuscript more easily read.

Author Response

(The authors gave the same response as above.)

Round 2

Reviewer 2 Report

 The Submission has been greatly improved and is worthy of publication.